# Influences of Glimepiride Self-Nanoemulsifying Drug Delivery System Loaded Liquisolid Tablets on the Hypoglycemic Activity and Pancreatic Histopathological Changes in Streptozotocin-Induced Hyperglycemic Rats

**DOI:** 10.3390/nano12223966

**Published:** 2022-11-10

**Authors:** Tarek A. Ahmed, Hanadi A. Alotaibi, Alshaimaa M. Almehmady, Martin K. Safo, Khalid M. El-Say

**Affiliations:** 1Department of Pharmaceutics, Faculty of Pharmacy, King Abdulaziz University, Jeddah 21589, Saudi Arabia; 2Department of Medicinal Chemistry, The Institute for Structural Biology, Drug Discovery and Development, School of Pharmacy, Virginia Commonwealth University, Richmond, VA 23298, USA

**Keywords:** glimepiride, SNEDDS, liquisolid tablets, industrial development, hypoglycemic activity, histopathology

## Abstract

The development of an oral anti-diabetic medication characterized by enhanced hypoglycemic activity is in high demand. The goal was to study the hypoglycemic activity and pancreatic histopathology after the black-seed-based self-nanoemulsifying drug delivery system (SNEDDS) loaded with glimepiride liquisolid tablets to diabetic rats. The solubility of glimepiride in various vehicles was investigated. An optimization SNEDDS formulation was developed using a mixture of the experimental design approach. Box–Behnken design (BBD) was used to develop glimepiride liquisolid tablets utilizing Avicel PH 101 and Neusilin as a carrier mixture and FujiSil as a coating material. The quality attributes of the prepared tablets were assessed. Following the administration of the optimized tablets to diabetic rats, the pharmacodynamics and histopathological changes were investigated and compared to a commercial drug product. Results revealed that the optimized SNEDDS formulation that contains 15.43% *w*/*w* black seed oil, 40% *w*/*w* Tween 80, and 44.57% *w*/*w* Polyethylene glycol 400 showed an average droplet size of 34.64 ± 2.01 nm and a drug load of 36.67 ± 3.13 mg/mL. The optimized tablet formulation contained 0.31% Avicel in the carrier mixture, a 14.99 excipient ratio, and 8% superdisintegrant. Pre- and post-compression properties were satisfactory, and the optimized glimepiride liquisolid tablet showed a two-fold increase in dissolution. The optimized tablet demonstrated superior pharmacodynamics. The pancreatic tissues of the group treated with the optimized tablet displayed normal histological structure. The obtained data offered a commercially viable alternative for manufacturing solid dosage forms containing water-insoluble drugs, but additional clinical research is required.

## 1. Introduction

The global population with diabetes grew four-fold from 108 million in 1980 to almost 422 million in 2014. In addition, the prevalence among adults increased to 8.5% in 2014, thus indicating an increase in associated risk factors such as being obese or overweight [1]. During the past few decades, the prevalence has dramatically expanded in both low- and middle-income countries [2]. Over the 16 years from 2000 to 2016, there was a rise of nearly 5% in the early death rate due to diabetes [3]. According to epidemiological modeling, the prevalence of type 2 diabetes mellitus (DM) in Saudi Arabia increased from 8.5% in 1992 to 39.5% in 2022 [4]. The mainstay of managing diabetes is proper diet and daily exercise, but when lifestyle modifications fail, treatment should be initiated [5]. Numerous anti-diabetic drugs are available with specific mechanisms of action to treat different types of diabetes. To choose the most appropriate treatment plan, knowing if the patient has insulin resistance, insulin deficiency, or both is necessary. Anti-diabetic drugs are broadly divided into non-insulin drugs such as but not limited to: insulin sensitizers (thiazolidinediones), insulin secretagogues (sulfonylureas and glinides), which are used for type 2 diabetes, and insulin analogs which are mainly used for type 1 diabetes [6]. Other oral anti-diabetic “ non-insulin” drugs act by suppressing the excessive hepatic glucose release and enhancing the hepatic insulin sensitivity, but this class has only a little impact on peripheral insulin-mediated glucose uptake (biguanides) [7]. One more class of oral anti-diabetic drugs acts as competitive, reversible inhibitors of pancreatic α-amylase and membrane-bound intestinal α-glucosidase inhibitors (Acarbose) [6]. Moreover, sodium-glucose co-transporter-2 inhibitors (Canagliflozin, Dapagliflozin, Empagliflozin, and Ertugliflozin) inhibit the reabsorption of the majority of the filtered glucose (approximately 90%) entering the kidney tubules [6]. The combined therapy of oral anti-diabetics, such as metformin and sulfonylureas or metformin with thiazolidinedione, has improved the care of patients with diabetes. It can be used if monotherapy is failed [5]. Despite many anti-diabetic agents in the market, sulfonylureas (SUs) are considered the most widely used in type 2 diabetes and have been utilized since 1950. SUs act mainly by raising the plasma insulin concentrations, especially when residual pancreatic β-cells are present, and this increment happens consequently due to inhibition of adenosine triphosphate (ATP)-sensitive potassium channels, which leads to cell depolarization and insulin exocytosis as well as suppression of hepatic clearance of insulin [8,9]. Glimepiride is a long-acting anti-diabetic drug that belongs to SUs. It enhances the intracellular insulin receptors’ activity and suppresses their down-regulation during chronic insulin stimulation through a mechanism involving protein kinase C activation. Glimepiride is practically insoluble in water and has an irregular dissolution and absorption profile, resulting in inconsistencies in bioavailability and drug action.

Self-nanoemulsifying drug delivery system (SNEDDS) is a homogenous multi-component system made up of oil, surfactant, co-surfactant, and a drug that can instantly dissolve to form fine nanoemulsion particles of nanometric range less than 200 nm upon mild agitation in an aqueous medium. After drug administration, the agitation required to form nanoemulsion is provided by the gastrointestinal tract (GIT) motility [10,11]. Due to self-emulsification in the GIT, the drug is presented in tiny oil droplets that enhance the dissolution profile by providing a large interfacial surface area for the drug to partition between the oil and the GIT fluid [12,13]. Contrary to ready-to-use suspensions and emulsions, SNEDDS is considered a thermodynamically stable formulation [12]. Natural or synthetic oil represents one of the main components of SNEDDS as it significantly helps in the spontaneity of the emulsification process, plays a role in nanoemulsion droplet size, transports the lipophilic drug through the intestinal lymphatic system, and accordingly enhances its GIT absorption [11,13]. Numerous oils showed anti-diabetic activity, including, but not limited to, argan, avocado, black seed, cactus pear, cinnamon, coconut, flaxseed, olive, rice bran, and soybean oils [14].

The liquisolid technique is used to modify the solubility and dissolution and consequently improve the bioavailability of poorly water-soluble drugs [15]. This technique is practically designed to formulate a flowable and easily compressible powdered form of a liquid medication which was first patented by Spireas [16]. The liquid medication can be an oily liquid, solution, or suspension of a drug carried in a suitable non-volatile solvent, often named a liquid vehicle [16,17]. The solvent should be compatible with the drug of interest and give the highest solubility. Three main components should be used during liquisolid formulation: liquid medication, carrier, and coating materials. Other ingredients, such as disintegrant or release retardant for manipulating the drug release profile, may be added according to the objective of the formulation [15]. Water solubility and wettability of liquisolid granules increased due to the addition of disintegrants such as croscarmellose sodium and crospovidone [18,19,20]. Conversion of a liquid SNEDDS into a solid SNEDDS using a liquisolid compact is a beneficial technological innovation as it merges the advantages of SNEDDS and oral solid dosage forms [21]. These two approaches (SNEDDS and liquisolid) have been employed together to improve SNEDDS stability in the GIT, enhance the dissolution, increase drug bioavailability, and solve the poor solubility and extensive first-pass metabolism [21,22,23,24].

In our recently published research, we have studied the pharmacokinetic behavior of glimepiride liquisolid tablets loaded with black seed oil-based SNEDDS and compared results to 3D-printed and directly compressed tablets [25]. In this work, we investigated the hypoglycemic activity and histopathological changes after administration of the prepared liquisolid tablets to diabetic rats. Additionally, more details about the experimental design, the studied ingredients, and the characterization of the prepared SNEDDS and liquisolid tablets are also mentioned.

## 2. Materials and Methods

### 2.1. Materials

Glimepiride was a kind gift from SPIMACO (Alqassim, KSA). Crosslinked Croscarmellose Sodium (Ac-Di-Sol^®^) was provided by BIOSYNTH^®^ Carbosynth International, Inc. (San Diego, CA, USA). Avicel^®^ PH 101 was brought from Fluka (Hach Lange, Ireland). Avocado oil was purchased from La Tourangelle (Woodland, CA, USA), and coconut oil from Nurture Vitality (Richmond, CA, USA). Black seed oil (100% pure cold-pressed) was obtained from Amazing Herbs (Atlanta, GA, USA). Flaxseed oil (cold-pressed) was purchased from Solgar (Leonia, NJ, USA). Cactus pear seed oil was a kind gift from a local supplier (Jeddah, KSA). Capryol^®^ PGMC, Labrafil^®^ M 1944CS, Labrafil^®^ M 2125CS, and Labrasol^®^ were obtained from Gattefossé (Lyon, France). Methanol, Kolliphor^®^ EL, Span^®^ 20, Span^®^ 85, and Tween^®^ 40 were brought from Sigma Aldrich Corp. (St. Louis, MO, USA). FujiSil^™^ and Neusilin^®^ US2 were purchased from Fuji Chemical Industries Co., Ltd. (Toyama, Japan). Glycerol was brought from Techno Pharmchem (Haryana, India). Magnesium stearate was brought from Winlab Laboratory Chemicals and Reagents (Leicestershire, UK). Talc powder was obtained from Whittaker Clark & Daniels (South Plainfield, NJ, USA). Tween^®^ 20, Tween^®^ 80, and Polyethylene glycol 400 (PEG 400) were obtained from Acros Organics (Geel, Belgium). Sodium lauryl sulfate (95% pure) was brought from Scharlau (Barcelona, Spain). Phosphoric acid and potassium dihydrogen phosphate were purchased from Riedel-de Haën AG (Seelze, Germany). Polyethylene glycol 200 (PEG 200) was brought from Merck Schuchardt (Hohenbrunn, Germany). Propylene glycol (PG) from Fluka (Steinheim, Germany). Acetonitrile HPLC grade was purchased from Merck (Darmstadt, Germany). All chemicals and solvents were of analytical grade.

### 2.2. Solubility Studies of Glimepiride in Different Vehicles

The solubility of glimepiride was evaluated in different oils, surfactants, and co-surfactants. Five different oils with anti-diabetic activity, namely: avocado oil, black seed oil, cactus pear seed oil, coconut oil, and flaxseed oil, were studied. The selected surfactants were Capryol^®^ PGMC, Kolliphor^®^ EL, Labrafil^®^ M 1944CS, Labrafil^®^ M 2125CS, Labrasol^®^, Span^®^ 20, Span^®^ 85, Tween^®^ 20, Tween^®^ 40, and Tween^®^ 80. The studied co-surfactants were glycerol, PEG 200, PEG 400, and PG. A known volume of 3 mL of each vehicle was placed in a screw-capped glass vial, and an excess amount of glimepiride was added to each vial. The prepared vials were mixed properly by a vortex mixer (Velp Scientifica, ZX3) and kept in a temperature-controlled shaking water bath (GFL Corporation, type 1083, Germany) at 25 ± 0.5 °C for 72 h. The content of each vial was filtered using a 0.45 μm syringe, diluted with methanol, and subjected to spectrophotometric analysis to determine its glimepiride content. All experiments were performed in triplicate, and results are expressed in mg/mL as the mean ± standard deviation (SD).

### 2.3. Development of SNEDDS Using Mixture Design

Due to the ability of the experimental design methods to develop nanoemulsion formulations of specific characteristics, the mixture design was used to develop optimized glimepiride-loaded SNEDDS formulation utilizing the Design-Expert^®^ version 13.0.9.0 (Stat-Ease Inc., Minneapolis, MN, USA) Software. Based on the solubility of glimepiride in the studied vehicles, ternary mixtures with different ratios of black seed oil (X1), Tween^®^ 80 (X2), and PEG 400 (X3) were used to prepare SNEDDS formulations. The particle size (Y1) and glimepiride solubility in the prepared SNEDDS formulations (Y2) were studied as responses (dependent variables). The studied range for black seed oil and Tween^®^ 80 was 10–40% *w*/*w*, while that for PEG 400 was 40–80% *w*/*w*. These ranges were selected based on our previous work for similar drug-loaded SNEDDS formulations [26,27]. Each of the three components was used in different ratios with a total mixture concentration of 100% to develop SNEDDS. A number of seventeen runs were prepared, as tabulated in Table 1.

### 2.4. Characterization of the Prepared SNEDDS Formulations

The prepared SNEDDS formulations were characterized for particle size, polydispersity index (PDI), zeta potential, and the highest drug solubility (maximum drug load). Briefly, 1 g of each SNEDDS formulation was added separately to a beaker containing 20 mL of double-distilled water and gently agitated over a magnetic stirrer. The obtained nanoemulsion formulations were assessed for particle size, polydispersity index (PDI), and zeta potential using Malvern Zetasizer Nano ZSP (Malvern Panalytical Ltd. (Malvern, UK)). The size was measured using dynamic light scattering with a non-invasive backscatter optics technique. An average of three measurements was recorded, and the results are expressed as the mean ± SD.

In order to select the SNEDDS formulation with the highest drug load, the solubility of glimepiride in the prepared SNEDDS formulations was studied. Briefly, an excess amount of glimepiride was added to 1 mL of each formulation in an Eppendorf safe-lock tube and vortexed until homogeneity. The tubes were kept in a temperature-controlled shaking water bath at 25 ± 0.5 °C for 72 h. Finally, the content in each tube was filtered using a 0.45 μm syringe filter, and the amount of glimepiride in the filtrate was assayed spectrophotometrically at 228 nm. All analyses were performed at least three times, and results are expressed in mg/mL as the mean ± SD.

### 2.5. Optimization of the Prepared SNEDDS

To develop an optimized SNEDDS formulation with the possible smallest droplet size and maximum drug load, regression equations were employed to correlate the relationship between the selected components and dependent variables using the Design-Expert^®^ version 13.0.9.0 (Stat-Ease Inc., Minneapolis, MN, USA) software.

The optimum SNEDDS formulation containing the proposed amounts of X1, X2, and X3 was prepared and characterized as described above, and the differences between the predicted and observed values were calculated.

### 2.6. Development of Glimepiride Self-Nanoemulsifying Liquisolid Tablets

#### 2.6.1. Preliminary Studies

Preliminary studies were used to select the suitable carrier and coating material that efficiently accommodates the drug-loaded SNEDDS formulation and produces dry-looking, free-flowing, and compressible powder mixture. Different carriers and coating materials were examined using the commonly used excipient ratio (R) and liquid load factor [19,28,29,30,31,32]. Neusilin^®^ and FujiSil^™^ were examined as a carrier and a coating material, respectively, in an excipient ratio of 10–15. Moreover, Fujicalin^®^ (as a carrier) and silica (as a coating material) were examined in an excipient ratio of 6.7 to 20. Finally, Avicel^®^ PH-101 (as a carrier) and silica (as coating material) were tested in an excipient ratio of 8 to 15.
(1)$$\mathrm{Excipient}\;\mathrm{ratio}\;\left(R\right)=\frac{\mathrm{Weight}\;\mathrm{of}\;\mathrm{carrier}\;(Q)}{\mathrm{Weight}\;\mathrm{of}\;\mathrm{coating}\;\mathrm{material}\;(q)}$$

#### 2.6.2. Design of Glimepiride Liquisolid Tablets Using Box–Behnken Design

Box–Behnken design (BBD), a type of response surface methodology, is useful for establishing a relationship between the selected independent variables and their responses [33]. This study used a three-factor, three-level BBD to prepare glimepiride liquisolid tablets utilizing the StatGraphics Centurion XV version 15.2.05 software, StatPoint Technologies, Inc. (Warrenton, VA, USA). The studied independent variables were the percentage of Avicel in the carrier mixture (X1), the excipient ratio (X2), and the percentage of superdisintegrant (X3). The studied levels for these variables were 0–50%, 5–15%, and 4–8% for X1, X2, and X3, respectively, which were chosen based on the preliminary study’s findings and the literature review. The impact of these variables on the pre- and post-compression characteristics of the powder mixtures was investigated. The angle of repose (Y1), tablet hardness (Y2), and the percentage of drug release after 30 min (Y3) were chosen as dependent variables. The goal was to minimize Y1 and maximize Y2 and Y2. A number of fifteen runs were suggested by the software, and their composition is tabulated in Table 2.

#### 2.6.3. Preparation of Glimepiride Liquisolid Tablets

Fifteen glimepiride liquisolid tablet formulations were prepared to utilize Avicel^®^ PH 101 and Neusilin^®^ as carriers, FujiSil^™^ as a coating material and Ac-Di-Sol^®^ in a concentration range of 4–8% *w*/*w* as a superdisintegrant. Moreover, Talc and magnesium stearate were added as a glidant and a lubricant, respectively.

The calculated amount of the carrier materials (Neusilin^®^ and Avicel^®^ PH 101) were accurately weighed using Mettler Toledo AJ100 electric balance (Greifensee, Switzerland) and blended in a mortar and pestle. The predetermined amount of GLIMEPIRIDE-loaded SNEDDS formulation was added, and mixing was continued until the homogenous distribution of the liquid medication was in the powder blend. The mixture was left standing for almost 5 min to allow better adsorption of the liquid medication on the powder surface. The coating material (FujiSil^™^) was added and blended well with the above mixture until a dry liquisolid powder was obtained. The superdisintegrant, glidant, and lubricant were subsequently added and thoroughly mixed. Finally, each run’s powder mixture was compressed into a round tablet containing 4 mg glimepiride (*n* = 30) using a double-punch tablet press machine of Erweka GmbH, AR 402 (Heusenstamm, Germany).

#### 2.6.4. Pre-Compression Characteristics

To enable efficient compression of the prepared liquisolid formulations, pre-compression characteristics are critical for achieving a powder blend with acceptable flowability standards. The flow properties of the prepared powder mixtures were determined by measuring the angle of repose (AOR), bulk and tapped densities, Carr’s index, and Hausner ratio. AOR was calculated through the fixed funnel method and using the following equation:(2)$$\tan\theta =\frac{\;h\;}{r}$$
where “h” and “r” represent the height and radius in centimeters (cm) of the formed cone, respectively.

Bulk density (BD), which is the weight-to-volume ratio of a powder mixture without tapping, was measured by placing a given weight of the powder in a graduated cylinder and using the obtained untapped volume to calculate the BD using the following equation:(3)$$\mathrm{BD}=\frac{\mathrm{Weight}\;\mathrm{of}\;\mathrm{the}\;\mathrm{powder}\;\mathrm{mixture}\;\left(W\right)}{\mathrm{Bulk}\;\mathrm{volume}\;\mathrm{of}\;\mathrm{the}\;\mathrm{same}\;\mathrm{powder}\;\mathrm{mixture}\;\left(\mathrm{Vb}\right)}$$

To estimate the tapped density (TD), the measuring cylinder was mechanically tapped ten times to obtain the tapped volume. The tapped density was calculated using the equation below:(4)$$\mathrm{TD}=\frac{\mathrm{Weight}\;\mathrm{of}\;\mathrm{the}\;\mathrm{powder}\;\mathrm{mixture}\;\left(W\right)}{\mathrm{Tapped}\;\mathrm{volume}\;\mathrm{of}\;\mathrm{the}\;\mathrm{same}\;\mathrm{powder}\;\mathrm{mixture}\;\left(\mathrm{Vt}\right)}$$

Bulk and tapped densities were utilized to evaluate the compressibility of the powder mixture by calculating the Hausner ratio and Carr’s index (C.I. %) according to the following equations:(5)$$C.I.\%=\left[\frac{\left(\mathrm{TD}-\mathrm{BD}\right)}{\mathrm{TD}}\right]\times 100$$
(6)$$\mathrm{Hausner}\;\mathrm{ratio}=\frac{\mathrm{TD}}{\mathrm{BD}}$$

#### 2.6.5. Post-Compression Characteristics

Weight variation was estimated by properly weighing individual tablets from each run (*n* = 20) using Mettler Toledo AJ100 electric balance (Greifensee, Switzerland). The results were expressed as an average weight in milligrams ± SD. Ten tablets from each run were randomly selected and individually measured for thickness using a Mitutoyo Co. (Kawasaki, Japan) digital micrometer. The obtained results are expressed as a mean in mm ± SD.

The force needed to break the tablet (*n* = 5) is measured using tablet hardness tester of Erweka GmbH, TBH 210 (Heusenstamm, Germany) to ensure that the prepared tablets are hard enough to resist breaking during handling and shipping and that they can properly disintegrate after swallowing. The average force in Newton (N) ± SD was used to express the results.

Ten tablets from each batch were taken and crushed individually in a mortar to ensure uniformity of the drug content. Methanol was added with continuous mixing until reaching a final volume of 100 mL. The resulting mixture was filtered using filter paper before being analyzed for glimepiride content using UV-VIS spectrophotometer at 228 nm. The data are expressed in percent (%) as mean ± SD.

The prepared tablets’ friability (*n* = 10) was assessed using an Erweka Friabilator type PTF1 from Pharmatest (Hainburg, Germany). After allowing the investigated tablets to rotate in the test device for 4 min at 25 rpm, friability was computed as a fraction of the weight of the original using the following equation:(7)$$\mathrm{Friability}\;\%=\frac{\left(\mathrm{Initial}\;\mathrm{weight}-\mathrm{final}\;\mathrm{weight}\right)}{\mathrm{Initial}\;\mathrm{weight}}\times 100$$

##### In Vitro Disintegration

Six tablets from each run were randomly selected and evaluated for in vitro disintegration at 37 °C in distilled water using Pharmatest’s USP34 tablet disintegration test instrument (Hainburg, Germany). The in vitro disintegration time of each tablet was determined after it was totally disintegrated, and no tablet fragments remained in the mesh of the basket.

##### In Vitro Release

Drug release from glimepiride liquisolid tablets was studied using the USP dissolution test apparatus type II (Pharma Test, Germany). The test was carried out in 500 mL double-distilled water containing 0.5% sodium lauryl sulfate at 37 ± 0.5 °C and 75 rpm. Samples of 5 mL were withdrawn at 15, 30, 45, 60, 90, and 120 min with immediate replacement with the same volume to maintain the sink condition. The withdrawn samples were analyzed spectrophotometrically at 228 nm. The experiment was carried out in triplicate, and results are expressed in % as the mean ± SD.

#### 2.6.6. Optimization of Glimepiride Liquisolid Tablet

The effect of X1, X2, and X3 on Y1, Y2, and Y3 was analyzed using the experimental design software. Regression equations for the relationship between independent and dependent variables were applied to identify the optimized formulation that achieves the study goal. Analyzed data were considered significant for any factor at a *p*-value less than 0.05. The optimized glimepiride liquisolid tablet was prepared and fully characterized, as mentioned above. Non-SNEDDS glimepiride tablets (without SNEDDS formulation) were also prepared by direct compression using the same excipients and the same optimized level of the studied variables. This formulation will be used in the subsequent investigation for comparison purposes.

### 2.7. Compatibility Studies

Fourier transform infrared (FT-IR) spectrum of glimepiride, Neusilin, Avicel^®^ PH101, FujiSil^™^, Ac-Di-Sol^®^, and the optimized liquisolid tablet formulation were investigated using a Nicolet Is10 FT-IR spectrometer from Thermo Scientific, Inc. (Waltham, MA, USA).

### 2.8. In Vivo Studies

Using a single glimepiride dose one-period parallel design, the pharmacokinetics of glimepiride from the prepared liquisolid, non-SNEDDS drug tablets, and commercial drug tablets were investigated. Male Wistar rats weighing 200–250 g were split into three groups, each with six animals. The non-SNEDDS glimepiride tablet “directly compressed tablet” formulation was given to Group I. The SNEDDS-loaded glimepiride tablet “liquisolid tablet” formulation was given to Group II. The commercial “marketed” drug tablets were given to Group III. Diabetes was induced two weeks before animal treatment by an intraperitoneal injection of 50 mg/kg streptozotocin. Accu-Chek Go (Roche, Mannheim, Germany) was used to measure fasting blood glucose levels. Animals with fasting blood glucose levels of 200–300 mg/100 mL were used. The rats were kept in a temperature-controlled closed enclosure with a 12-h light/dark cycle for one week prior to treatment. Animals have unrestricted access to food and water. The dose of glimepiride utilized was 10 mg/kg [25].

As previously studied with similar liquisolid [31] and orodispersible tablets [34], each tablet formulation was crushed and suspended in one-percent carboxymethyl cellulose to facilitate ease of administration of the formulation to the animals through a stomach tube easier. At 0.5, 1, 2, 4, 6, 8, 12, and 24 h, blood samples (0.25 mL) were taken from the animals via retro-orbital puncturing under light ether anesthesia. The plasma samples were immediately separated and kept at 20°C after centrifugation at 6000 rpm for 10 min. The study was carried out following Good Clinical Practice, the International Conference on Harmonization, and the European Medicines Agency recommendations. The Research Ethics Committee of the Faculty of Pharmacy at King Abdulaziz University in Saudi Arabia has already approved the study’s protocol (Reference No. 1021442).

#### 2.8.1. Chromatographic Conditions

Calibration standards were made from glimepiride and metformin (internal standard) methanolic stock solutions to determine glimepiride levels in plasma samples. Using a Perkin Elmer high-performance liquid chromatograph (HPLC) with variable wavelength UV detector and autosampler, the concentration of glimepiride in the unknown and the prepared calibration standards was determined. Phenomenex RP Hi-Q-Sil C18 column (250 × 4.6 mm, 5 m, Phenomenex Inc., Torrance, CA, USA) was used for separation. The mobile phase was made up of 64% acetonitrile and 36% potassium dihydrogen orthophosphate (0.02 M), pH 3.5 adjusted. The mobile phase flow rate was adjusted to 1 mL/min. The wavelength of the UV detector was set at 230 nm. 1 mL of acetonitrile-methanol (1:1) mixture was added to each plasma sample for preparation and extraction. The mixture was vortexed for 1 min before being centrifuged for 10 min at 5000 rpm. Under a steady stream of nitrogen at 50 °C, the organic phase was separated and entirely evaporated to dryness. The residue was reconstituted in 80 μL of mobile phase with a 30 μL injection volume. Except for minor changes, the HPLC method for the measurement of glimepiride in plasma samples was replicated [25].

#### 2.8.2. Pharmacokinetic Analysis

Following oral administration of the optimized glimepiride tablet formulations and the commercial tablets, a non-compartmental extravascular pharmacokinetic model utilizing PKsolver (an add-in program for pharmacokinetic data) was used to determine the following pharmacokinetic parameters. The maximum (peak) plasma concentration over the defined time period (C_max_) and the time point at which the maximum plasma concentration was reached (T_max_) were also specified. The linear trapezoidal method was used to estimate the area under the plasma concentration–time curve from zero time to the last measurable concentration (AUC0–t), and the area under the plasma concentration–time curve from time zero to infinity (AUC0–inf) was calculated as the sum of the AUC0–t plus the ratio of the last measurable plasma concentration to the elimination rate constant.

### 2.9. Hypoglycemic Activity

The hypoglycemic activity of the developed tablet formulations was examined on the studied diabetic rats to investigate the pharmacodynamic activity. Animals were classified into three groups and given the same treatment as previously mentioned in the pharmacokinetics study. A control group (normal rats) and animals with untreated induced hyperglycemia (model) were also used. The hypoglycemic activity of glimepiride following an oral treatment was evaluated after 0.5, 1, 2, 4, 6, 8, 12, and 24 h. The percent reduction in blood glucose level was also calculated.

### 2.10. Histopathological Assessment of the Pancreas

The pancreas tissues of the animals used in the pharmacodynamics study were obtained and processed with 10% neutral buffer formalin after the animals were sacrificed. Using a microtome, paraffin slices were cut to the necessary thickness and stained with hematoxylin and eosin. Finally, any changes in the pancreas tissues were examined under a microscope. The area of the Langerhans islets was assessed microscopically in the analyzed pancreas specimens.

### 2.11. Statistical Analysis

The pharmacokinetic data of glimepiride were represented as mean ± SD and were statistically evaluated using the GraphPad Prism 8 software (GraphPad Inc., La Jolla, CA, USA). A two-way ANOVA with Tukey’s multiple comparisons test was used to compare each group’s means at all-time points to determine the significance of the difference. Results with a significance level of *p*-value < 0.05 were considered significant.

## 3. Results and Discussion

### 3.1. Determination of Glimepiride Solubility in Different Vehicles

Glimepiride is a drug with a low water solubility of 1.6 μg/mL [35]. The solubility of glimepiride in various vehicles was investigated to determine the best oil, surfactant, and co-surfactant for developing SNEDDS with a high drug load. Figure 1 illustrates the obtained results. Black seed oil showed the highest glimepiride solubility among the studied oils. Moreover, Tween^®^ 80 and PEG 400 were chosen as surfactant and co-surfactant, respectively.

### 3.2. Characterization of the Prepared SNEDDS

The average particle size, PDI, and zeta potential are essential parameters in SNEDDS formulations as they monitor the stability of the formulations. In this study, the mixture design was implemented to develop an SNEDDS formulation with the smallest particle size and highest drug load. The mean particle size for the prepared SNEDDS formulations was in the range of 32.12 ± 5.30 to 82.89 ± 5.76 nm, while the PDI ranged from 0.310 ± 0.028 to 0.709 ± 0.112. The zeta potential values ranged from −0.3267 ± 0.2199 mV to 0.2332 ± 0.7025 mV. The drug load was varied from 14.59 ± 0.91 mg/mL to 40.07 ± 2.05 mg/mL. Table 1 shows the results for the size and drug load. Results of the particle size analysis revealed that the prepared SNEDDS formulations showed PDI values between 0.313 ± 0.018 and 0.646 ± 0.188 which is an indication of an appropriate particle size distribution. Danaei et al. found that particles with a PDI greater than 0.7 have an extremely broad size distribution [36]. Because the studied components, black seed oil, Tween**^®^** 80, and PEG 400, are non-ionic, the zeta potential values revealed almost neutral “uncharged” particles. More discussion about the size and drug load of the prepared SNEDDS formulation will be explained in the next section.

### 3.3. Statistical Analysis for Optimization of Glimepiride-Loaded SNEDDS

The mixture experimental design was used to develop an optimized SNEDDS formulation utilizing black seed oil, Tween^®^ 80, and PEG 400 as oil, surfactant, and co-surfactant, respectively. The obtained results for the size and drug solubility were statistically analyzed using the Design-Expert^®^ version 13.0.9.0 (Stat-Ease Inc., Minneapolis, MN, USA) Software.

Results of the particle size indicated that a smaller SNEDDS size of 32.12 nm (run 7) and 39.09 nm (run 14) was produced at low oil and high surfactant percent. On the contrary, when low surfactant and high oil percent were used, a large SNEDDS size of 82.89 nm (run 2), 79.42 nm (run 15), and 81.56 nm (run 1) was obtained. Fitting the data to different models indicated that the Cubic model is the suggested one, as illustrated in the model summary statistics represented in Table 3. Analysis of variance (ANOVA) for the Cubic model of particle size indicated that a model F-value of 72.31 was obtained, which implies that the model is significant. There is only a 0.01% chance that an F-value this large could occur due to noise. Moreover, the calculated *p*-value was found to be less than 0.0001, which confirms that the model was significant. *p*-values less than 0.05 indicate model terms are significant. Values greater than 0.1 indicate the model terms are not significant. The calculated Lack of Fit F-value of 1.17 implies that the Lack of Fit is not significant relative to the pure error. There is a 38.31% chance that a Lack of Fit F-value this large could occur due to noise. Non-significant lack of fit is good since we want the model to fit. The actual equation in terms of actual components is:Y1 = 5.65132 X1 − 13.21639 X2 + 2.04124 X3 + 0.291570 X1X2 + 0.116371 X1X3 + 0.201714 X2X3 − 0.001438 X1X2X3 − 0.001301 X1X2X1X2 + 0.001578 X1X3X1X3 + 0.001494 X2X3X2X3(8)

The equation in terms of actual factors can be used to make predictions about the response for given levels of each factor. Here, the levels should be specified in the original units for each factor. This equation should not be used to determine the relative impact of each factor because the coefficients are scaled to accommodate the units of each factor, and the intercept is not at the center of the design space. A contour plot for the effect of X1, X2, and X3 on Y1 and Y2 was constructed and is illustrated in Figure 2.

Fitting the drug solubility results to different models illustrated that the linear model is the suggested one as indicated from the values of the standard deviation, R-squared, Adjusted R-squared, and Predicted R-squared displayed in Table 3. ANOVA for the linear model of the drug solubility demonstrated that a model F-value of 87.61 was obtained, which implies that the model is significant. The calculated *p*-value was less than 0.0001. Moreover, a Lack of Fit F-value of 1.54 was obtained, which means that the Lack of Fit is insignificant relative to the pure error. There is a 33.06% chance that a Lack of Fit F-value this large could occur due to noise. Again, a non-significant lack of fit is good since we want the model to fit. The equation of the drug solubility in terms of actual components is:Y2 = 0.240729 X1 + 0.814958 X2 + 0.087125 X3(9)

Greater drug solubility was obtained at a high surfactant percent of 40% (run 5 and 17), followed by 34.63% (runs 7 and 14), and finally at 29.92% (run 12). This effect could be attributed to the better drug solubility in Tween**^®^** 80. Glimepiride demonstrated a solubility of 70.336 mg/mL in Tween**^®^** 80, as previously mentioned in Figure 1. Accordingly, a higher percent of Tween**^®^** 80 in the prepared SNEDDS formulation resulted in greater drug solubility.

### 3.4. Preparation and Characterization of the Optimized Glimepiride-Loaded SNEDDS

An optimized formulation was predicted, prepared, and characterized as above-stated about the fitted regression models. The optimum levels for X1, X2, and X3 were found to be 15.43, 40, and 44.57, respectively. The predicted values for Y1 and Y2 were 31.10 nm and 40.19 %, respectively, while the observed values were 34.64 nm and 36.67 %, respectively. Residual values (difference between predicted and observed) that are less than 4% were obtained.

### 3.5. Development of Glimepiride Liquisolid Tablet

Determination of the flowability of the liquisolid powder mixture helps identify the most suitable liquisolid excipients that can be compressed into tablets. Based on the preliminary studies, there was a marked difference in the powder blends’ liquid adsorption capacity and flow properties among the studied carriers and coating materials. Formulations with Neusilin^®^ and FujiSil^™^ mixture at the excipient ratio between 10 and 15 exhibited good to excellent liquid adsorption capacity and flow properties. In contrast, Fujicalin^®^ and silica produced wet powder with poor flow properties even at a higher excipient ratio of 20. In addition, using Avicel^®^ PH-101 and silica resulted in a dry powder blend at an excipient ratio of 15. Therefore, due to the variations in the physical appearance and properties of the powder blend, it was found that the most suitable liquisolid carrier to be chosen in this study together with glimepiride-loaded SNEDDS is Neusilin^®^ and/or Avicel^®^ PH-101. Moreover, FujiSil^™^ is the preferable coating material in comparison with silica.

#### 3.5.1. Evaluation of the Prepared Liquisolid Powder Blend

Table 2 displays the pre-compression characteristics of the prepared glimepiride liquisolid powder mixture. The bulk and tapped densities for all the fifteen liquisolid powder blends (LS-1 to LS-15) were estimated, and the obtained results were used to calculate the Hausner ratio and Carr’s index. The calculated values of the angle of repose varied from 28.39° to 38.00°, which indicates acceptable results. Moreover, the obtained values for the Carr’s index and Hausner ratio ranged between 8.70 to 23.68% and from 1.10 to 1.31, respectively. These results revealed acceptable flow properties of the powder mixture of all the suggested runs according to the United States Pharmacopeia (USP) powder flow <1174> [37].

#### 3.5.2. Quality Attributes of the Prepared Glimepiride Liquisolid Tablets

The compressed tablets were visually inspected and assessed for quality parameters such as weight uniformity, content uniformity, hardness, friability, and in vitro disintegration after effective compression of the glimepiride liquisolid powder mixture formulations. Visual examination confirmed the absence of sticking and picking. The punches were clean and free from any defects. Table 4 displays the obtained result of the post-compression characteristics. The prepared tablets showed acceptable weight variation and were found to meet USP requirements since no more than two tablets deviated from the average weight by more than 5%, and none deviated by more than 10% [38]. The friability of the tablets ranged from 0.007% to 0.164%, which is within the acceptable USP range (less than 1%). Tablet hardness ranged between 46 ± 3.61 N to 110 ± 5.13 N, and the thickness of tablets was between 4.07 ± 0.02 mm and 4.47 ± 0.04 mm. The mean in vitro disintegration time for all the studied tablets was less than 30 min which met the USP requirement [38]. The LS-9 formulation illustrated the shortest in vitro disintegration time of 3.78 ± 1.19 min, while the longest time (9.12 ± 0.59 min) was observed with the LS-5 formulation. The percentage of Avicel in the carrier mixture (X1) and the superdisintegrant percent (X3) significantly impacted the in vitro disintegration time. As demonstrated in LS-5, a high percentage of Avicel in the carrier mixture and a low percentage of superdisintegrant resulted in a longer disintegration. In addition, as demonstrated in LS-9, a low percent of Avicel in the carrier mixture and a high superdisintegrant percent resulted in a shorter time. It has previously been mentioned that Avicel is one of the commonly used pharmaceutical tablet excipients with the best binding properties [39]. It promotes dry component blending, resulting in tablets with high hardness and minimal friability, as well as excellent compression [40]. Moreover, Neusilin, an amorphous magnesium aluminometasilicate, has been incorporated in tableting to produce hard tablets at low compression force [41]. Croscarmellose sodium (Ac-Di-Sol) is the most compatible superdisintegrant with Neusilin. The large surface area and porous nature of Neusilin and the cross-linking characteristic of Croscarmellose sodium act synergistically to allow the tablet to swell and absorb its weight in water many times, resulting in rapid disintegration [42]. Accordingly, increasing the Avicel concentration in the carrier mixture will result in a lower Neusilin percentage in the carrier, the effect that is expected to increase the disintegration time. Moreover, the incorporation of a high percent of Ac-Di-Sol results in excellent water uptake and rapid swelling properties, resulting in a faster in vitro disintegration time. The content uniformity of the prepared tablets ranged from 93.85 ± 0.56% to 104.37 ± 8.94%, which met the standards of Pharmacopeia, which stipulates that the range of contents must be between 85% and 115%. It is noteworthy to mention that there are methods for reflection spectroscopy which could help to monitor tablet fabrication for quality control [43].

#### 3.5.3. In Vitro Release

The dissolution studies of the prepared glimepiride liquisolid tablets were performed in double-distilled water containing 0.5% sodium lauryl sulfate as a dissolution medium to maintain the sink condition. The in vitro release profile of glimepiride from the fifteen runs is demonstrated in Figure 3. The prepared liquisolid formulations were observed to release 62.75 to 102.34% of glimepiride within the first 30 min and 75.15 to 109.47% within the first hour. Most studied formulations exhibited a fast in vitro drug release during the first 30 min. This behavior could be attributed to the rapid tablet disintegration (3.78 ± 1.19 to 9.12 ± 0.59 min) and the presence of the drug in the nano-sized form (SNEDDS). The positive effect of the oil and the emulsifier on the in vitro drug release has been previously reported for the release of griseofulvin from drug-loaded lyophilized dry emulsion tablets [44]. It should be mentioned that the formulation factors had a considerable impact on the glimepiride release profile from the prepared liquisolid tablets, which will be discussed in depth in the following section.

#### 3.5.4. Statistical Analysis for Optimization of Glimepiride Liquisolid Tablets

Box–Behnken design was employed to develop the optimized glimepiride liquisolid tablets formulation. The data obtained for the studied responses (Y1, Y2, and Y3) were statistically analyzed using multiple response optimization and ANOVA using Statgraphics software. The observed and fitted values for Y1, Y2, and Y3 are illustrated in Table 5. In addition, the values for the estimated effects of the studied factors, F-ratios, and the corresponding *p*-values obtained from the ANOVA are demonstrated in Table 6. A positive value for the estimated effect implies that this independent variable synergistically influences the examined response. The F-ratio compares the actual and expected variations in the variable averages; an F-ratio larger than 1 indicates the presence of a location impact, and hence the *p*-value is used to indicate the level of significance. If the *p*-value differs from 0 and is less than 0.05, a factor is regarded to have a significant effect on the studied response. In contrast, a negative value suggests that it has an antagonistic effect.

The equations of the fit model are:Y1 = 0.868 + 0.247 X1 + 2.184 X2 + 4.955 X3 − 0.0001 X1^2^ − 0.003 X1X2 − 0.021 X1X3 − 0.034 X2^2^ − 0.168 X2X3 − 0.183 X3^2^(10)
Y2 = 56.836 + 2.183 X1 + 2.317 X2 − 8.043 X3 − 0.019 X1^2^ − 0.115 X1X2 − 0.128 X1X3 + 0.135 X2^2^ − 0.092 X2X3 + 1.136 X3^2^(11)
Y3 = 39.829 − 0.079 X1 + 1.371 X2 + 3.289 X3 + 0.012 X1^2^ − 0.0006 X1X2 − 0.009 X1X3 + 0.073 X2^2^ − 0.313 X2X3 + 0.206 X3^2^(12)

##### Effect of the Independent Variables on Angle of Repose (Y1)

The obtained values of the angle of repose for the prepared liquisolid formulations varied from 28.39° to 38.00°, as shown in Table 5, indicating that all the formulations exhibited excellent to fair flow. It was found that there is a direct relationship between all the studied variables (X1, X2, and X3) on the angle of repose (Y1), as illustrated in the Pareto chart of Figure 4. The chart shows a reference line with a *p*-value of 0.05. Any factor effect that surpasses this line has a significant impact on the response being evaluated. Formulation LS-1 demonstrated the highest angle of repose (38.00°) due to the presence of the high level of X1 (percentage of Avicel^®^: 50%) and the high level of X2 (excipient ratio: 15). On the contrary, formulation LS-3 showed the lowest angle of repose (28.39°) due to its low level of X1 and X2. Three-dimensional response surface plots were created to describe the effect of varying two independent variables on a studied response while keeping the third variable at its intermediate level. Figure 4 illustrates these charts graphically.

As the percent of Avicel in the carrier mixture was increased, the degree of the angle of repose was increased, and hence the flow was rated poor. This behavior could be attributed to the presence of a low amount of Neusilin at a high percent of X1, which is characterized by its large surface area and porous nature [42]. Moreover, the excipient ratio (X2), the ratio between the weights of carrier and coating materials in the formulation, exhibited a positive effect on Y1. As the excipient ratio was increased, the angle of repose was increased, and poor flow was obtained. This effect could be attributed to the presence of a high level of the coating material (FujiSil^™^) at a low excipient ratio, which leads to the formation of a dry, free-flowing, and compressible powder mixture, as previously mentioned by Lu et al. [19]. An explanation for this behavior is that the drug-loaded SNEDDS is first absorbed into the carrier’s internal framework. Once the carrier’s interior has been saturated with liquid medication, a liquid layer forms on the carrier particles’ surface, which is quickly absorbed by the tiny coating materials [19]. Finally, the superdisintegrant demonstrated a positive effect on Y1, which indicates poor flowability at a high level of X3. The impact could be attributed to the interference of the superdisintegrant with the FujiSil^™^ action.

##### Effect of the Independent Variables on the Tablet Hardness (Y2)

It was found that the percent of Avicel in the carrier mixture (X1) and the excipient ratio (X2) were significantly affecting the tablet hardness (Y2), as illustrated in the Pareto chart Figure 5. ANOVA revealed that X1 had an antagonistic effect on the tablet hardness, while X2 had an agonistic (positive) effect. Figure 5 also shows three-dimensional estimated response surface plots, which validated this finding. Tablets having Neusilin alone as a carrier (LS-7) had a high hardness value, whereas those containing 50% Neusilin (LS-2) had a low hardness value. This finding is consistent with a prior study that found that tablet hardness declined in the sequence listed below: Neusilin alone > Neusilin + Corn Starch > Neusilin + Cross-link polyvinylpyrrolidone > Neusilin + carboxymethyl cellulose > Neusilin + rich starch [42]. Moreover, the hardness of a lactose tablet compounded with 10% Neusilin was higher than that of a tablet made with 15% microcrystalline cellulose [41]. The increase in the tablet hardness at a high excipient ratio (X2) could be attributed to the FujiSil^™^ content. According to a previous study, tablets made with Neusilin had a higher hardness than those made with FujiSil^™^ [45]. Accordingly, as the excipient ratio decreased from 15 to 5, the amount of the coating substance (FujiSil^™^) increased, and the amount of carrier (either Neusilin or Neusilin/Avicel) in the tablet decreased, resulting in less hard tablets.

##### Effect of the Independent Variables on the Drug Release (Y3)

Based on the Pareto chart depicted in Figure 6, it was noticed that the percent of Avicel in the carrier mixture (X1), excipient ratio (X2), and the superdisintegrant percent had a positive impact on the drug release (Y3). Figure 6 also represents the estimated response surface plots for the effects of the selected variables on the release of the drug (Y3). Three-dimensional response surface plots were constructed to investigate the effect of varying two independent variables while keeping the third variable at its intermediate level on the in vitro drug release after 30 min, as displayed in Figure 6. At a high percent of Avicel in the carrier mixture, the in vitro drug release was increased; the effect could be attributed to the low tablet content of Neusilin US2 that is characterized by its high specific surface area (300 m^2^/g), high oil adsorbing capacity (2.7–3.4 mL/g), and its high water adsorbing capacity (2.4–3.1 mL/g) [41], which may lead to strong physical interaction between the drug-loaded SNEDDS formulation and Neusilin, and so less drug releases. Similarly, as the excipient ratio was increased, the drug release was increased. This behavior could be attributed to the low amount of FujiSil^™^ at a high level of the excipient ratio. FujiSil^™^ is characterized by its high specific surface area (400 m^2^/g) and high oil adsorbing capacity (3.3 mL/g) and has been used for the development of granules with controlled release behavior [45]. As a result, tablets with a large amount of FujiSil^™^ are likely to have reduced drug release behavior. Finally, the drug release from the prepared tablets increased as the percent of superdisintegrant was raised. Ac-Di-Sol^®^ has been claimed to have a disintegrant effect by promoting water uptake into tablet pores and by acting as a water wick [46]. At high levels of Ac-Di-Sol^®^, this impact is likely to intensify. Kharshoum et al. reported short in vitro and in vivo disintegration times and maximum in vitro drug release for vinpocetine orally disintegrating tablets at a high level of Ac-Di-Sol^®^ [47].

##### Preparation and Characterization of the Optimized Glimepiride Liquisolid Tablets

Based on the fitted regression models, the proposed optimized glimepiride liquisolid tablets formulation was prepared and fully characterized as previously described. The optimized level for X1, X2, and X3 was found to be 0.31%, 14.99, and 8.0%, respectively. The observed values for the angle of repose (Y1), tablet hardness (Y2), and drug release (Y3) were found to be 32.38, 114.67, and 73.38, respectively. Our results are in good agreement with previous work, which stated that combining a self-emulsifying drug delivery strategy with liquisolid technology can be a promising tool for improving the dissolution profile of leflunomide in vitro [30]. Moreover, drug-loaded SNEDDS could also be loaded into mucoadhesive systems to modify the drug release in the gastrointestinal tract [48].

### 3.6. Compatibility Studies

Figure 7 illustrates the FT-IR spectra of the studied samples. The carbonyl (C = O) stretching of glimepiride has two distinct peaks which were identified at 1705.82 and 1672.36 cm^−1^. At 1347.83 cm^−1^, another drug peak for C–N stretching vibrations was discovered. At 1156.66 cm^−1^, the sulfoxide (S = O) stretching vibration was detected. The FT-IR spectrum of the optimized liquisolid tablets formulation demonstrated the disappearance of the drug characteristic peaks. Internalization of the drug particles in the SNEDDS formulation before developing the liquisolid tablets could account for this effect. Chavda et al. reported a similar finding for isotretinoin-loaded solid SNEDDS formulation. They reported that none of the isotretinoin’s distinctive peaks were found in the solid SNEEDS formulation, and authors attributed this finding to the fact that the drug is in a molecularly dissolved condition in the SNEDDS formulation [49]. The same result was previously mentioned for three-dimensional printed tablets containing Rosuvastatin and Glimepiride-loaded Curcuma oil SNEDDS [50].

### 3.7. In Vivo Study

After a single oral glimepiride treatment of 10 mg/kg to induced-hyperglycemic male Wistar rats, plasma concentration–time profiles for the SNEDDS-loaded glimepiride tablets were compared to non-SNEDDS-loaded glimepiride and marketed tablets, as illustrated in Figure 8. The two-way ANOVA revealed a significant difference (*p*-value < 0.05) between the studied groups at most of the time points. Table 7 summarizes the computed values of the pharmacokinetic parameters of glimepiride. The addition of glimepiride to the SNEDDS formulation hastens the absorption of the drug, allowing it to reach its maximum plasma concentration (C_max_) in 2 h rather than 4 h as with non-SNEDDS-loaded tablets. The C_max_ of glimepiride for SNEDDS-loaded tablets was improved more than two times compared to non-SNEDDS-loaded tablets. In addition, compared to non-SNEDDS glimepiride tablets, SNEDDS-loaded glimepiride tablets had a much higher area under the curve, which was reflected in a considerable improvement in relative bioavailability. The marketed tablets showed a 2 h T_max_, a C_max_ of 2158.67 ± 128.81 ng/mL, an elimination half-life of 5.94 ± 0.03 h, a 15,366.83 ± 597.84 ng/mL*h area under the plasma level time curve from zero time to the last measurable concentration, and a mean residence time of 9.14 ± 0.09 h. The obtained results suggest that the incorporation of glimepiride in black seed oil-based SNEDDS with minimum globule size, followed by loading onto liquisolid tablets, leads to an increase in the rate and extent of absorption as well as improving the oral bioavailability of glimepiride which is in agreement with the previously reported results [31,51,52,53,54].

### 3.8. Hypoglycemic Activity

As shown in Figure 9, the blood glucose level in the diabetic rats (model group) increased significantly (*p* < 0.05) when compared to the normal rats (control group). This finding demonstrates that the studied rats developed diabetes after receiving a 50 mg/kg intraperitoneal injection of streptozotocin two weeks before glimepiride treatment. There was a fall in blood glucose level following administration of non-SNEDDS glimepiride tablets, although it only lasted for about 4 h. The hypoglycemic effectiveness of SNEDDS-loaded glimepiride tablets was significantly reduced (*p* < 0.05) after 1 h and extended to 8 h compared to the model group. The percentage reduction in blood glucose level after administration of the examined tablets is depicted in Figure 10. Compared to the model group, SNEDDS-loaded glimepiride tablets revealed a 40 percent reduction after 1 h, a 60 percent reduction after 4 h, and a 40 percent reduction after 8 h. On the other hand, non-SNEDDS loaded glimepiride tablets only exhibited a 30 to 40% reduction in blood glucose levels within the first 6 h. This suggests that the SNEDDS-loaded formulations have a significantly longer duration of effect than the non-SNEDDS-loaded formulations. The marketed glimepiride tablets exhibited a significant decrease in blood glucose levels and a significant reduction in blood glucose level percentage.

The results obtained indicate that SNEDDS-loaded glimepiride tablets outperform the non-SNEDDS and marketed drug tablets in terms of lowering blood glucose levels and percent reduction in blood glucose. However, neither impact was as strong as the SNEDDS-loaded glimepiride tablets. The previous study has shown that transdermal patches loaded with glimepiride SNEDDS increased the drug skin permeability, bioavailability, and hypoglycemic activity when compared to a marketed product [55]. Moreover, Rangaraj et al. referred to the development of Febuxostat SNEDDS in liquid SNEDDS, solid SNEDDS, and pellet formulations. These formulations were superior to pure Febuxostat in their oral bioavailability and anti-gout activity, indicating that the SNEDDS formulation could be considered an effective gout treatment alternative [56]. El-Say et al. reported improved oral bioavailability, hypolipidemic, and cardioprotective efficacy of simvastatin from drug-loaded liquisolid tablets compared to commercial tablets and attributed this behavior to better drug disintegration and absorption [31]. Moreover, Kazi et al. showed improved in vivo performance of Talinolol SNEDDS, which was attributed to an increase in the drug solubility and dissolution at pH 1.2 and 6.8 from the prepared SNEDDS, which is more likely to improve absorption in the GIT [57].

### 3.9. Histopathological Investigation of the Pancreas

The normal pancreatic anatomy of both exocrine units and endocrine components was demonstrated by microscopic examination of pancreas tissues from the control group (Figure 11A). Langerhans islets looked to be of normal size and contained β-cells. The model group’s pancreas tissues (Figure 11B) showed necrosis of islets of Langerhans admixed with the accumulation of eosinophilic tissue debris (black arrow). The pancreatic parenchyma was moderately protected when the animals were given the marketed tablets. It was typical to see degraded endocrine islets with vacuolar degeneration cells.

In some cases, necrotic islets were also discovered. A focal region of significant hemorrhage was discovered (Figure 11C). The islets of the non-SNEDDS group’s Langerhans were degenerated, atrophied, and necrotic, with vacuolated cells and profoundly eosinophilic necrotic cells. There were also a lot of clogged vessels (Figure 11D). Pancreas tissue from the animal group administered with the optimized SNEDDS glimepiride tablet appeared to have a normal structure in several analyzed sections. The endocrine components revealed normal large variable-sized Langerhans islets with intact β-cells. A significant number of clogged blood arteries were found between the exocrine acini in a few areas (Figure 11E,F).

## 4. Conclusions

A successful combination of black seed oil, Tween^®^ 80, and PEG 400 was used to develop an optimized SNEDDS with a small globule size and high drug loading capacity. Box–Behnken design was implemented to develop glimepiride loaded-SNEDDS liquisolid tablets utilizing Neusilin^®^, FujiSil^™^, and Avicel^®^ PH 101 as carrier and coating materials. The developed tablet formulations demonstrated acceptable quality control characteristics and improved glimepiride dissolution profile. The optimized drug-loaded SNEDDS liquisolid tablets formulation demonstrated superior pharmacokinetic behavior that was accompanied by a significant improvement in the hypoglycemic activity when compared to pure drug and marketed tablets. Histopathological examination of the pancreatic tissues of diabetic rats treated with the prepared liquisolid tablets revealed a normal structure. These findings confirm that the conversion of a drug-loaded liquid SNEDDS into a solid form using a liquisolid compact is a promising approach as it merges the advantages of SNEDDS and oral solid dosage forms.

## Figures and Tables

**Figure 1 nanomaterials-12-03966-f001:**
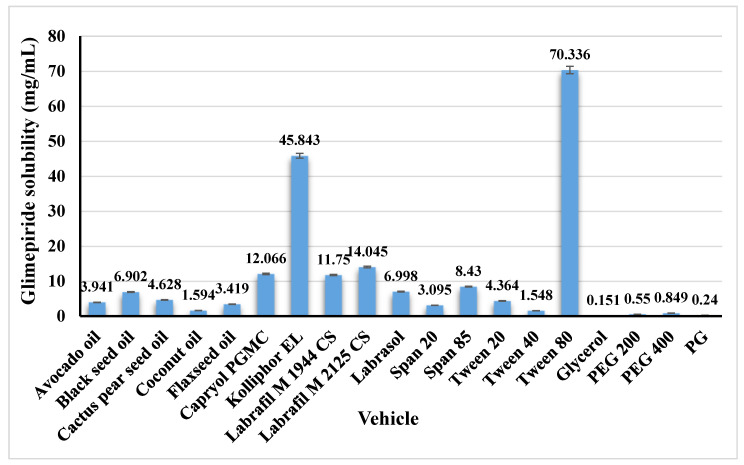
Solubility of glimepiride in the studied vehicles. Data are presented as mean ± SD (*n* = 3).

**Figure 2 nanomaterials-12-03966-f002:**
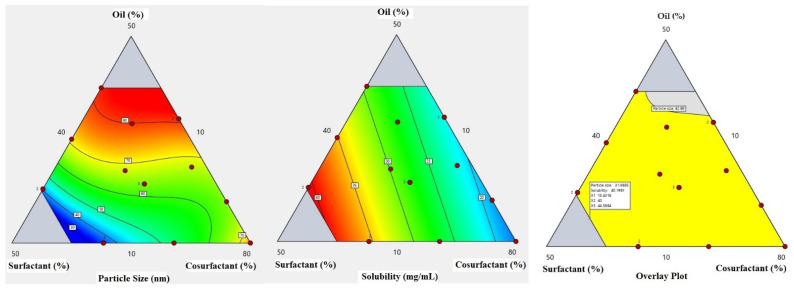
Contours for the particle size and drug solubility and overlay plot for the effect of the studied factors on particle size and solubility.

**Figure 3 nanomaterials-12-03966-f003:**
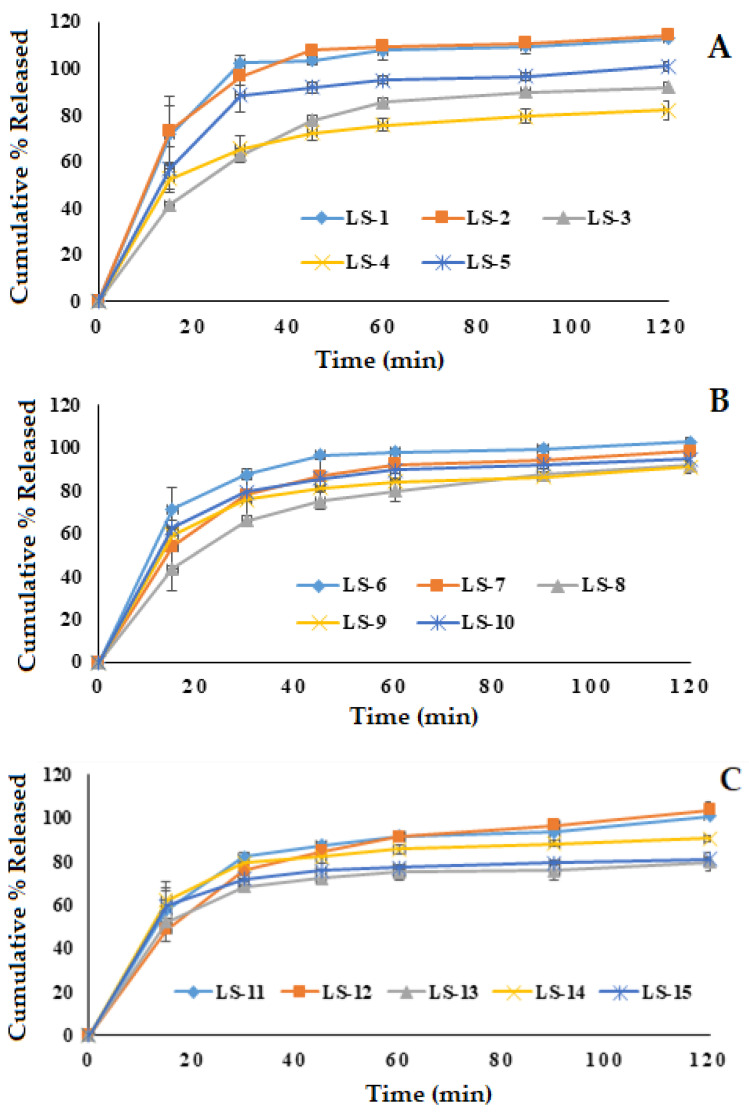
In vitro release of glimepiride from the prepared fifteen liquisolid tablet formulations. (**A**); LS-1: LS-5, (**B**); LS-6: LS-10, (**C**); LS-11: LS-15.

**Figure 4 nanomaterials-12-03966-f004:**
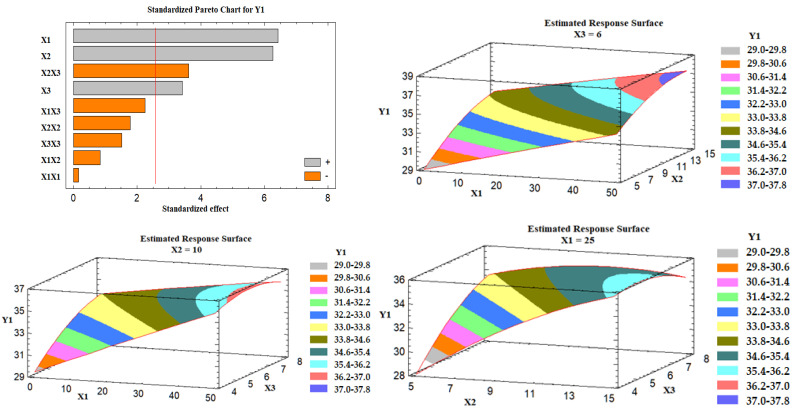
Pareto chart and estimated response surface plots for the effect of the studied factors on the angle of repose (Y1). Notes: X1; the carrier mixture, X2; the excipient ratio; X3; the percentage of superdisintegrant.

**Figure 5 nanomaterials-12-03966-f005:**
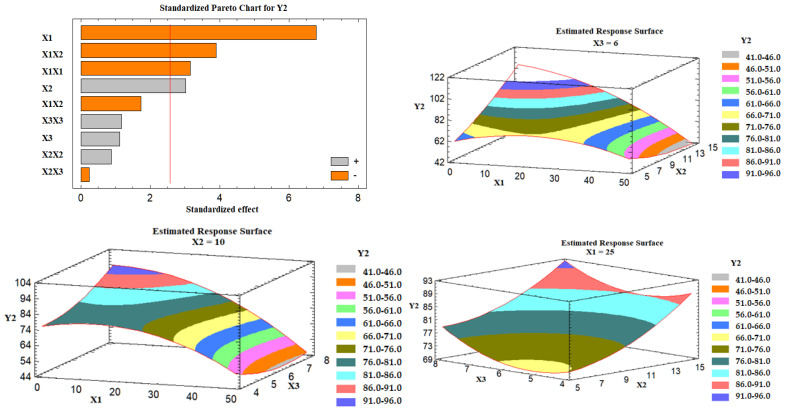
Pareto chart and estimated response surface plots for the effect of the studied factors on the tablet hardness (Y2). Notes: X1; the carrier mixture, X2; the excipient ratio; X3; the percentage of superdisintegrant.

**Figure 6 nanomaterials-12-03966-f006:**
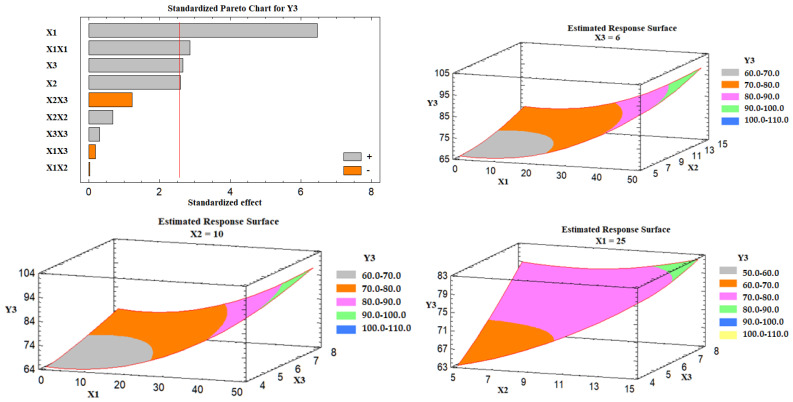
Pareto chart and estimated response surface plots for the effect of the studied factors on the drug release (Y3). Notes: X1; the carrier mixture, X2; the excipient ratio; X3; the percentage of superdisintegrant.

**Figure 7 nanomaterials-12-03966-f007:**
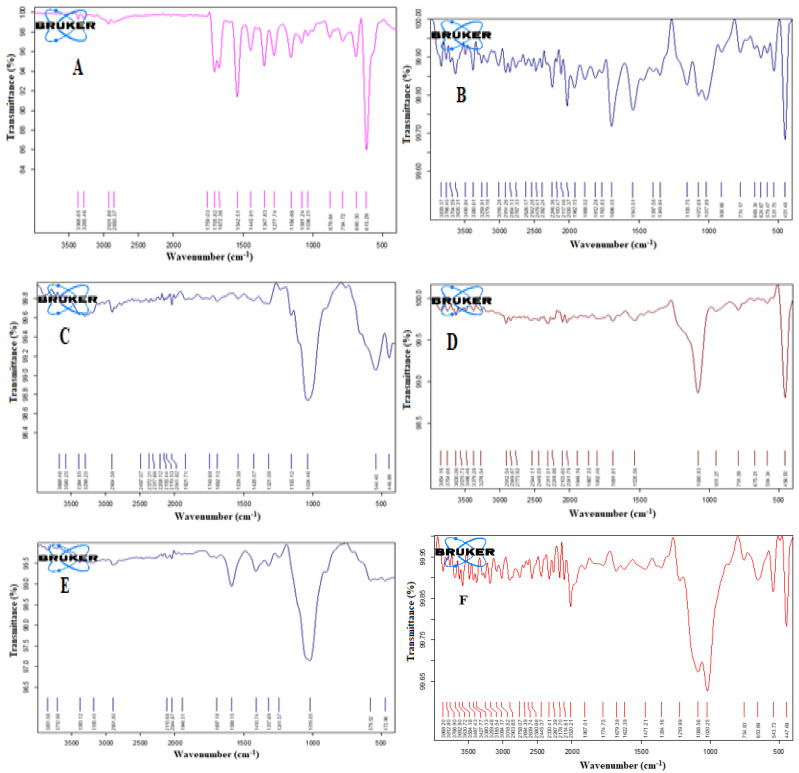
FT-IR spectra of Glimepiride (**A**), Neusilin (**B**), Avicel (**C**), FujiSil (**D**), Ac-Di-Sol (**E**), and the optimized liquisolid tablet (**F**).

**Figure 8 nanomaterials-12-03966-f008:**
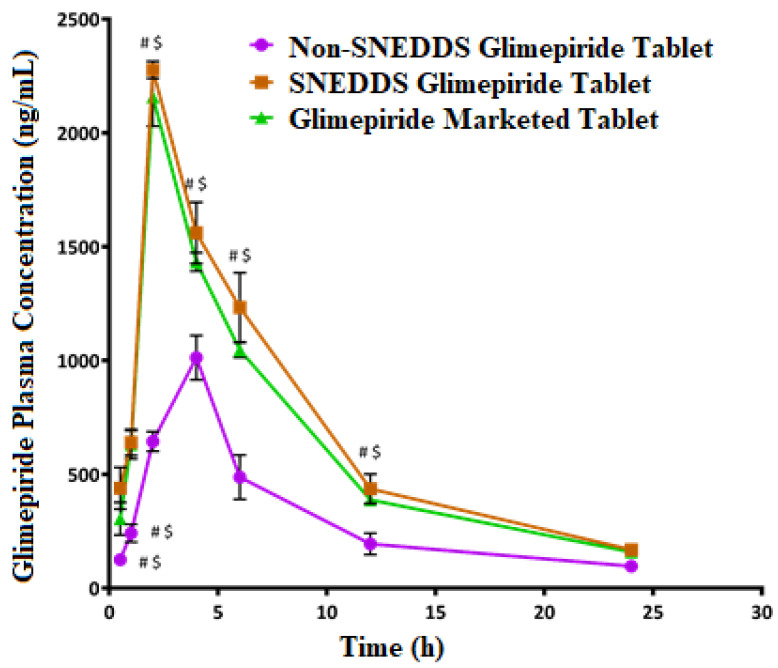
Plasma concentration–time curves of glimepiride after oral administration of the SNEDDS, non-SNEDDS, and marketed tablets to induced-hyperglycemic male Wistar rats (*n* = 6). # and $ Indicate significant difference between the Non-SNEDDS and marketed tablets, respectively, at *p*-value < 0.05.

**Figure 9 nanomaterials-12-03966-f009:**
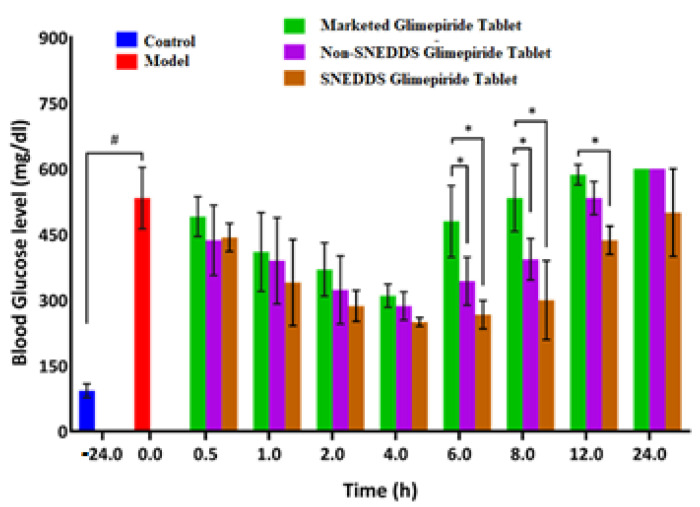
Blood glucose level after oral administration of the SNEDDS, non-SNEDDS, and marketed glimepiride tablets to induced-hyperglycemic male Wistar rats (*n* = 6). Notes: # Indicates significance from the control group, * Indicates significance from marketed tablets. Significant difference was considered at *p* < 0.05. Data are presented as mean ± SD, (*n* = 3).

**Figure 10 nanomaterials-12-03966-f010:**
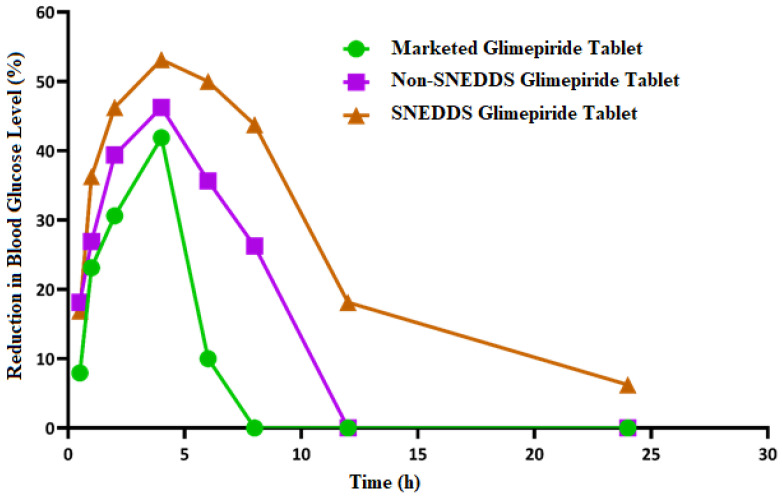
Inhibition of blood glucose (%) after oral administration of the SNEDDS, non-SNEDDS, and marketed glimepiride tablets to induced-hyperglycemic male Wistar rats (*n* = 6).

**Figure 11 nanomaterials-12-03966-f011:**
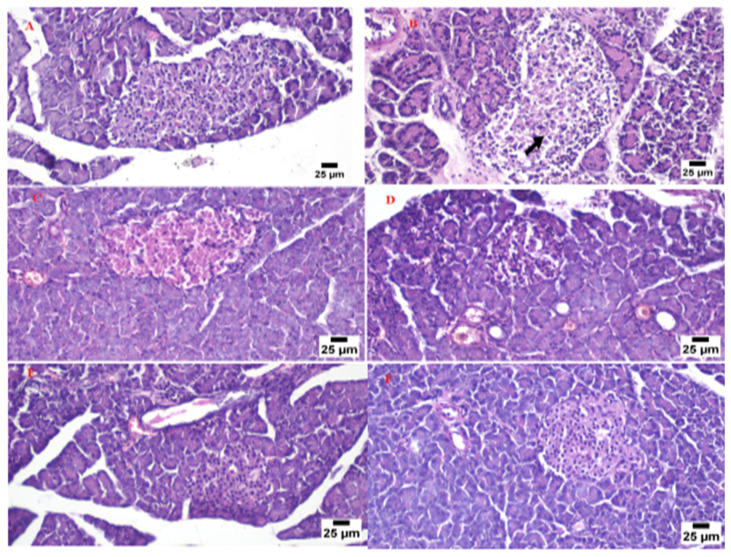
Photomicrograph for the pancreas tissue of the control group (**A**), model group (**B**), marketed product (**C**), non-SNEDDS glimepiride tablet (**D**), and SNEDDS glimepiride tablet (**E**,**F**).

**Table 1 nanomaterials-12-03966-t001:** Composition and characterization of SNEDDS formulations according to the ternary mixture design.

Run Code	Black Seed Oil (%)	Tween^®^ 80 (%)	PEG 400 (%)	Particle Size (nm)		Drug Solubility (mg/mL)	
				Mean	SD	Mean	SD
F1	40.00	20.00	40.00	81.56	5.56	27.27	1.46
F2	34.02	10.00	55.98	82.89	5.76	21.52	1.06
F3	10.00	22.80	67.20	51.78	2.74	29.41	1.33
F4	17.98	10.00	72.02	62.37	4.29	19.14	1.04
F5	20.00	40.00	40.00	40.68	2.55	40.07	2.05
F6	33.09	18.29	48.62	78.83	2.18	25.07	1.73
F7	10.00	34.63	55.37	32.12	5.30	34.29	2.69
F8	10.00	10.00	80.00	75.32	4.89	14.59	0.91
F9	24.69	12.50	62.81	68.59	4.13	19.07	1.30
F10	24.00	24.00	52.00	65.55	3.31	30.28	2.03
F11	21.43	22.05	56.52	64.52	6.90	28.14	1.16
F12	30.08	29.92	40.00	71.86	2.62	35.89	2.06
F13	21.43	22.05	56.52	64.52	6.90	28.14	1.16
F14	10.00	34.63	55.37	39.09	3.64	37.47	1.25
F15	34.02	10.00	55.98	79.42	4.66	26.45	1.20
F16	21.43	22.05	56.52	64.52	6.90	28.14	1.16
F17	20.00	40.00	40.00	40.68	2.48	40.07	3.05

**Table 2 nanomaterials-12-03966-t002:** Composition of glimepiride liquisolid formulations as suggested by the Box–Behnken design and their pre-compression characteristics.

Run Code	Avicel in Carrier Mixture (%)	Excipient (Ratio)	Superdisintegrant (%)	Angle of Repose (°)	Densities (mg/mL)		Carr’s Index %	Hausner Ratio
					Bulk	Tapped		
LS-1	50	15	6	38.00	277.29	363.34	23.68	1.31
LS-2	50	10	8	36.22	272.71	330.55	17.50	1.21
LS-3	0	5	6	28.39	232.05	269.83	14.00	1.16
LS-4	0	10	4	29.12	225.00	246.42	8.70	1.10
LS-5	50	10	4	35.11	269.08	316.56	15.00	1.18
LS-6	50	5	6	33.69	277.37	323.60	14.29	1.17
LS-7	0	15	6	34.26	217.73	241.92	10.00	1.11
LS-8	25	5	4	29.06	257.18	282.90	9.09	1.10
LS-9	0	10	8	34.41	226.70	264.49	14.29	1.17
LS-10	25	15	8	33.46	251.96	296.05	14.89	1.18
LS-11	25	5	8	33.69	253.10	283.97	10.87	1.12
LS-12	25	15	4	35.54	235.38	270.69	13.04	1.15
LS-13	25	10	6	34.63	256.13	300.03	14.63	1.17
LS-14	25	10	6	34.88	263.43	308.59	14.63	1.17
LS-15	25	10	6	34.08	260.99	305.73	14.63	1.17

**Table 3 nanomaterials-12-03966-t003:** Model summary statistics for glimepiride-loaded SNEDDS on droplet size and drug solubility.

	Particle Size				Drug Solubility			
	SD	R^2^	Adj. R^2^	Pred. R^2^	SD	R^2^	Adj. R^2^	Pred. R^2^
Linear	5.89	0.884	0.867	0.797	2.15 *	0.926 *	0.915 *	0.877 *
Quadratic	4.29	0.952	0.929	0.732	2.16	0.942	0.915	0.842
Special Cubic	4.31	0.955	0.929	0.522	2.03	0.953	0.925	0.877
Cubic	2.52 *	0.989 *	0.976 *	0.675 *	1.74	0.976	0.945	0.495
Special Quartic	4.77	0.956	0.913	0.395	1.74	0.972	0.945	0.828
Quartic	2.46	0.993	0.976	-	1.86	0.980	0.937	-

SD: Standard deviation, R^2^: R-squared, Adj-R^2^: Adjusted R-squared, Pred. R^2^: Predicted R-squared. *: Focus of the model maximizing the Adjusted R² and the Predicted R².

**Table 4 nanomaterials-12-03966-t004:** Results of the quality control tests for the prepared glimepiride liquisolid tablets.

Run Code	Weight ^a^ (mg) Mean ± SD	Hardness ^b^ (N) Mean ± SD	Friability ^c^ (%)	Thickness ^c^ (mm) Mean ± SD	Disintegration ^d^ (min) Mean ± SD	Content Uniformity ^c^ (%) Mean ± SD
LS-1	393.14 ± 1.63	47 ± 1.15	0.020	4.21 ± 0.03	7.62 ± 0.50	101.90 ± 4.43
LS-2	406.76 ± 1.83	46 ± 3.61	0.164	4.31 ± 0.03	7.20 ± 0.07	96.83 ± 3.52
LS-3	414.84 ± 2.10	56.67 ± 5.51	0.092	4.13 ± 0.01	4.63 ± 0.54	103.95 ± 3.30
LS-4	391.72 ± 5.69	76 ± 4.36	0.059	4.22 ± 0.10	4.96 ± 0.53	93.85 ± 0.56
LS-5	391.03 ± 1.44	52 ± 1.73	0.007	4.37 ± 0.02	9.12 ± 0.59	104.37 ± 8.94
LS-6	414.62 ± 1.57	51.33 ± 8.62	0.067	4.47 ± 0.04	6.30 ± 0.83	99.79 ± 7.06
LS-7	393.28 ± 0.77	110 ± 5.13	0.020	4.07 ± 0.02	4.46 ± 0.67	100.96 ± 1.26
LS-8	407.88 ± 1.82	76 ± 2.08	0.099	4.29 ± 0.02	8.68 ± 0.63	101.34 ± 13.94
LS-9	406.33 ± 4.19	95.67 ± 8.01	0.039	4.12 ± 0.10	3.78 ± 1.19	104.74 ± 1.21
LS-10	400.84 ± 1.94	88 ± 1.69	0.038	4.13 ± 0.07	6.63 ± 0.40	97.06 ± 0.29
LS-11	422.54 ± 6.07	82.67 ± 5.95	0.042	4.15 ± 0.07	7.08 ± 0.84	99.52 ± 3.59
LS-12	385.70 ± 5.16	85 ± 0.71	0.040	4.13 ± 0.04	7.95 ± 0.59	99.62 ± 3.00
LS-13	398.16 ± 2.03	77.33 ± 2.10	0.051	4.14 ± 0.05	5.64 ± 0.51	103.14 ± 0.99
LS-14	398.91 ± 0.97	82.67 ± 4.85	0.045	4.11 ± 0.01	5.02 ± 1.05	102.38 ± 0.52
LS-15	398.13 ± 1.36	75 ± 3.17	0.038	4.18 ± 0.02	5.16 ± 1.71	100.11 ± 0.76

^a^: *n* = 20, ^b^: *n* = 5, ^c^: *n* = 10, ^d^: *n* = 6.

**Table 5 nanomaterials-12-03966-t005:** Observed and fitted values for the selected responses as per Box–Behnken design.

Run Code	Dependent Variables					
	Angle of Repose (°)		Hardness (N)		Release (%)	
	Observed	Fitted	Observed	Fitted	Observed	Fitted
LS-1	38.00	37.35	47.0	42.0	102.34	98.82
LS-2	36.22	35.90	46.0	46.17	96.68	97.56
LS-3	28.39	29.04	56.67	61.67	62.75	66.27
LS-4	29.12	29.45	76.0	75.83	65.62	64.74
LS-5	35.11	35.74	52.0	53.17	88.39	88.91
LS-6	33.69	34.03	51.33	54.99	87.52	89.64
LS-7	34.26	33.93	110.0	106.33	77.85	75.73
LS-8	29.06	28.08	76.0	71.17	65.92	63.28
LS-9	34.41	33.77	95.67	94.50	75.80	75.28
LS-10	33.46	34.44	88.0	92.83	79.55	82.19
LS-11	33.69	33.68	82.67	78.84	82.13	79.13
LS-12	35.54	35.55	85.0	88.83	75.87	78.87
LS-13	34.63	34.53	67.33	75.0	68.43	73.22
LS-14	34.88	34.53	82.67	75.0	79.29	73.22
LS-15	34.08	34.53	75.0	75.0	71.95	73.22

**Table 6 nanomaterials-12-03966-t006:** Estimated effects of factors, F-ratio, and associated *p*-value for glimepiride liquisolid tablets on the angle of repose, hardness, and drug release.

Factors	Angle of repose (°)			Hardness (N)			Release (%)		
	Estimated Effect	F-Ratio	*p*-Value	Estimated Effect	F-Ratio	*p*-Value	Estimated Effect	F-Ratio	*p*-Value
X1	0.2472	41.26	0.0014 *	2.1832	46.09	0.0011 *	−0.0790	41.77	0.0013 *
X2	2.1840	39.30	0.0015 *	2.3168	9.17	0.0292 *	1.3711	6.73	0.0486 *
X3	4.9548	11.68	0.0189 *	−8.0425	1.25	0.3152	3.2888	7.12	0.0444 *
X1X1	−0.0001	0.03	0.8726	−0.0194	9.93	0.0254 *	0.0121	8.20	0.0353 *
X1X2	−0.0031	0.71	0.4381	−0.1153	15.20	0.0114 *	−0.0006	0.00	0.9791
X1X3	−0.0209	5.08	0.0740	−0.1283	3.01	0.1432	−0.0095	0.03	0.8598
X2X2	−0.0344	3.18	0.1345	0.1350	0.77	0.4207	0.0727	0.47	0.5224
X2X3	−0.1678	13.11	0.0152 *	−0.0919	0.06	0.8139	−0.3133	1.52	0.2725
X3X3	−0.1829	2.30	0.1897	1.1356	1.39	0.2910	0.2065	0.10	0.7675
R^2^	95.8760			94.6111			92.9180		
Adj-R^2^	88.4529			84.9109			80.1703		
SEE	0.9267			7.3954			5.0823		
MAE	0.4499			3.5117			2.4992		

*: Indicates the significance of factors on each response individually at *p*-value < 0.05.

**Table 7 nanomaterials-12-03966-t007:** Pharmacokinetic parameters of glimepiride after oral administration of 10 mg/kg in rats (*n* = 6, the data expressed as mean ± SD).

Parameter	Non-SNEDDS Tablet	SNEDDS Tablet	Marketed Tablet
K (h^−1^)	0.09 ± 0.00	0.12 ± 0.00	0.12 ± 0.00
t½(h)	8.09 ± 0.22	5.90 ± 0.10	5.94 ± 0.03
Tmax (h)	4.00 ± 0.00	2.00 ± 0.00	2.00 ± 0.00
Cmax (ng/mL)	1013.33 ± 97.45	2277.00 ± 37.40	2158.67 ± 128.81
AUC 0-t (ng/mL*h)	7523.08 ± 1254.54	17,121.42 ± 1453.77	15,366.83 ± 597.84
AUC 0-inf (ng/mL*h)	8655.33 ± 1556.11	18,562.32 ± 1431.03	16,729.63 ± 632.37
AUMC 0-inf (ng/mL*h^2)	97,360.68 ± 22,926.65	167,942.91 ± 10,391.00	152,838.93 ± 5370.34
MRT 0-inf (h)	11.18 ± 0.60	9.05 ± 0.14	9.14 ± 0.09

## Data Availability

Not applicable.

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
