# Peer review of "Influences of Glimepiride Self-Nanoemulsifying Drug Delivery System Loaded Liquisolid Tablets on the Hypoglycemic Activity and Pancreatic Histopathological Changes in Streptozotocin-Induced Hyperglycemic Rats"

_nanomaterials, 2022, doi:10.3390/nano12223966_

Round 1
Reviewer 1 Report
The authors did a remarkable screening work with very interesting results. They did solid statistics and compare solubility, and also processability of a antidiabetic drug in several different oils, co-surfactants and detergents. The work is quite solid and there are only minor adaptions necessary prior to publishing.
1. References fort the diabetes prevalence missing. Please provide reference for each claim, also in the first few sentences.
2. I would like to know the references to the claim that oral antidiabetic drugs improve insulin sensitivity, and beta cell sensitivity.
3. There are lots of other claims which are not referenced.
4. What is an appropriate particle size distribution? Please specify the values.
5. Numbers in Figure 2 are hard to read, please use larger numbers
6. I suggest to add labels like a-c and describe what the different compartments in Figure 3 actually mean.
7. Pointed out peaks of the wavelengths are not readable, maybe increasing size helps.
8. The authors could mention, that there are methods for reflection spectroscopy[1] which could help to monitor tablet fabrication for quality control.
9. There are mucoadhesive systems,[2] which could be loaded with the system of the authors. They could state it as an outlook, for defined release in the gastrointestinal tract.
10. Figure 7: Please label the compartments a-d and mention in figure caption what they are.
11. The authors write about a black arrow in Figure 10 b but I did not see it, please make it more clear.
12. There is a competitive system which is also of small size and even sugar resceptive. The authors could point out the strength of their system.[3]
13. Reference 4 needs to have a DOI or publisher information and date of publishing or more information in general, currently not ok like this.
References
[1] A. Früh, S. Rutkowski, I.O. Akimchenko, S.I. Tverdokhlebov, J. Frueh, Orientation analysis of polymer thin films on metal surfaces via IR absorbance of the relative transition dipole moments, Appl Surf Sci. 594 (2022) 153476. https://doi.org/10.1016/j.apsusc.2022.153476.
[2] S. Rutkowski, T. Si, M. Gai, M. Sun, J. Frueh, Q. He, Magnetically-guided Hydrogel Capsule Motors produced via Ultrasound assisted Hydrodynamic Electrospray Ionization Jetting, J Colloid Interface Sci. 25 (2019) 752–774. https://doi.org/10.1016/j.jcis.2019.01.103.
[3] T. Levy, C. Déjugnat, G.B. Sukhorukov, Polymer Microcapsules with Carbohydrate-Sensitive Properties, Adv Funct Mater. 18 (2008) 1586–1594. https://doi.org/10.1002/adfm.200701291.
Author Response
Reviewer 1
The authors did a remarkable screening work with very interesting results. They did solid statistics and compare solubility, and also processability of a antidiabetic drug in several different oils, co-surfactants and detergents. The work is quite solid and there are only minor adaptions necessary prior to publishing.
- References fort the diabetes prevalence missing. Please provide reference for each claim, also in the first few sentences.
Reply
The requested references have been added to the revised manuscript. Kindly refer to references 1 and 2.
- I would like to know the references to the claim that oral antidiabetic drugs improve insulin sensitivity, and beta cell sensitivity.
Reply
The sentence has been modified and references have been cited in the modified manuscript.
- There are lots of other claims which are not referenced.
Reply
More references have been included in the revised manuscript.
- What is an appropriate particle size distribution? Please specify the values.
Reply
The requested values for the PDI have been added.
- Numbers in Figure 2 are hard to read, please use larger numbers.
Reply
A modified figure with clear numbers has been added to the revised manuscript.
- I suggest to add labels like a-c and describe what the different compartments in Figure 3 actually mean.
Reply
Labels (A, B and C) and description of the formulations have been added to figure 3.
- Pointed out peaks of the wavelengths are not readable, maybe increasing size helps.
Reply
All the manuscript figures have been submitted as a separate file in a clear mode.
- The authors could mention, that there are methods for reflection spectroscopy [1] which could help to monitor tablet fabrication for quality control.
Reply
The requested information and reference have been added. (see section 3.5.2.).
- There are mucoadhesive systems,[2] which could be loaded with the system of the authors. They could state it as an outlook, for defined release in the gastrointestinal tract.
Reply
The requested information and reference have been added. (See section 3.5.4.3.).
- Figure 7: Please label the compartments a-d and mention in figure caption what they are.
Reply
The requested modifications have been done.
- The authors write about a black arrow in Figure 10 b but I did not see it, please make it more clear.
Reply
The figure has been modified.
- There is a competitive system which is also of small size and even sugar resceptive. The authors could point out the strength of their system.[3]
Reply
The strength of the SNEDDS is illustrated in the introduction section, paragraph 2.
- Reference 4 needs to have a DOI or publisher information and date of publishing or more information in general, currently not ok like this.
Reply
Refernce has been corrected.
References
[1] A. Früh, S. Rutkowski, I.O. Akimchenko, S.I. Tverdokhlebov, J. Frueh, Orientation analysis of polymer thin films on metal surfaces via IR absorbance of the relative transition dipole moments, Appl Surf Sci. 594 (2022) 153476. https://doi.org/10.1016/j.apsusc.2022.153476.
[2] S. Rutkowski, T. Si, M. Gai, M. Sun, J. Frueh, Q. He, Magnetically-guided Hydrogel Capsule Motors produced via Ultrasound assisted Hydrodynamic Electrospray Ionization Jetting, J Colloid Interface Sci. 25 (2019) 752–774. https://doi.org/10.1016/j.jcis.2019.01.103.
[3] T. Levy, C. Déjugnat, G.B. Sukhorukov, Polymer Microcapsules with Carbohydrate-Sensitive Properties, Adv Funct Mater. 18 (2008) 1586–1594. https://doi.org/10.1002/adfm.200701291.

Reviewer 2 Report
The manuscript written by Ahmad et al. is devoted to the development of glimepiride self-nanoemulsifying drug delivery system loaded liquisolid tablets. The manuscript represent a complex study which includes the development of nanoformulations, their characterization and biological evaluation. In the general, it is well written and structured. However, du to the large volume of the paper some additional improvements are required to make it clearer. Please, find below my comments and suggestions:
1. Table 1-3, 5. In my opinion, it is better to avid the abbreviations in column headings such as Y1, Y2 etc. The certain parameters stay behind this abbreviations and they should be reflected in the column headings. First, it make the results clearer from the first glance. Second, it allows the readers to avoid misleadings since the same letters are used in different tables to indicate the different parameters. For example, Tables 4 and 7 are correctly prepared. 2. Figures 3 and 7 are multicomponent. However, no dividing to subfigures was done. I recommend to mark the graphs as a,b, c… and add the clarification to them into the figure legend. 3. Figures 4-6. Please explain in the figure legend what is Y1, Y2 and Y3. 4. Figure 10. Please add the error bars to each plot. I recommend to start the figure legend as: Inhibition of blood glucose (%) after … 5. Figure containing images of histological analysis in fact has number 11 but not 10. Please, correct as well as its mentioning in the text. Also, it would be better to mark the images, for example, to mark the cell types, clogged vessel, etc. to support the observations discussed in the text. 6. Conclusions are poor and need the revision. Some details, findings and tendencies should be underlined in this section.Author Response
Reviewer 2
The manuscript written by Ahmad et al. is devoted to the development of glimepiride self-nanoemulsifying drug delivery system loaded liquisolid tablets. The manuscript represent a complex study which includes the development of nanoformulations, their characterization and biological evaluation. In general, it is well written and structured. However, due to the large volume of the paper some additional improvements are required to make it clearer. Please, find below my comments and suggestions:
- Table 1-3, 5. In my opinion, it is better to avid the abbreviations in column headings such as Y1, Y2 etc. The certain parameters stay behind this abbreviations and they should be reflected in the column headings. First, it make the results clearer from the first glance. Second, it allows the readers to avoid misleadings since the same letters are used in different tables to indicate the different parameters. For example, Tables 4 and 7 are correctly prepared.
Reply
We highly appreciate the reviwer comment. Abbreviations in the column headings for tables 1-3, 5 have been replaced with the full name.
- Figures 3 and 7 are multicomponent. However, no dividing to subfigures was done. I recommend to mark the graphs as a,b, c… and add the clarification to them into the figure legend.
Reply
Figure 3 and 7 have been modified as suggested by the reviewer. Also, all the manuscript figures have been submitted as a separate file in a clear mode.
- Figures 4-6. Please explain in the figure legend what is Y1, Y2 and Y3.
Reply
Explanation for Y1, Y2 and Y3 has been added.
- Figure 10. Please add the error bars to each plot. I recommend to start the figure legend as: Inhibition of blood glucose (%) after …
Reply
The figure legend has been modified. Values illustrated in figure 10 represent the average % inhibition in blood glucose level between the studied tablets and the model group, accordingly there are no error bars.
- Figure containing images of histological analysis in fact has number 11 but not 10. Please, correct as well as its mentioning in the text. Also, it would be better to mark the images, for example, to mark the cell types, clogged vessel, etc. to support the observations discussed in the text.
Reply
The figure has been modified.
- Conclusions are poor and need the revision. Some details, findings and tendencies should be underlined in this section.
Reply
The conclusion section has been revised. More details and findings have been added.